# A Novel Tactile Function Assessment Using a Miniature Tactile Stimulator

**DOI:** 10.3390/s23041844

**Published:** 2023-02-07

**Authors:** Chung-Tung Sung, Yung-Jung Wang, Jian-Jia Huang, Yu-Cheng Pei, Lei-Chi Lin, Wen-Hsin Mai, Bao-Luen Chang

**Affiliations:** 1Department of Medical Education, Chang Gung Memorial Hospital at Linkou Medical Center, Taoyuan 33305, Taiwan; 2School of Medicine, College of Medicine, Chang Gung University, Taoyuan 33302, Taiwan; 3Department of Physical Medicine and Rehabilitation, Chang Gung Memorial Hospital at Linkou Medical Center, Taoyuan 33305, Taiwan; 4Center of Vascularized Tissue Allograft, Chang Gung Memorial Hospital at Linkou Medical Center, Taoyuan 33305, Taiwan; 5Master of Science Degree Program in Innovation for Smart Medicine, Chang Gung University, Taoyuan 33302, Taiwan; 6School of Chinese Medicine, College of Medicine, Chang Gung University, Taoyuan 33302, Taiwan; 7Department of Neurology, Chang Gung Memorial Hospital at Linkou Medical Center, Taoyuan 33305, Taiwan

**Keywords:** tactile, tactile acuity, tactile stimulator, robotic, miniature, automated, grating ball

## Abstract

Several methods for the measurement of tactile acuity have been devised previously, but unexpected nonspatial cues and intensive manual skill requirements compromise measurement accuracy. Therefore, we must urgently develop an automated, accurate, and noninvasive method for assessing tactile acuity. The present study develops a novel method applying a robotic tactile stimulator to automatically measure tactile acuity that comprises eye-opened, eye-closed training, and testing sessions. Healthy participants judge the orientation of a rotating grating ball presented on their index fingerpads in a two-alternative forced-choice task. A variable rotation speed of 5, 10, 40, or 160 mm/s was used for the tactile measurement at a variety of difficulties. All participants met the passing criteria for the training experiment. Performance in orientation identification, quantified by the proportion of trials with correct answers, differed across scanning directions, with the highest rotation speed (160 mm/s) having the worst performance. Accuracy did not differ between vertical and horizontal orientations. Our results demonstrated the utility of the pre-test training protocol and the functionality of the developed procedure for tactile acuity assessment. The novel protocol performed well when applied to the participants. Future studies will be conducted to apply this method to patients with impairment of light touch.

## 1. Introduction

Somatosensation, also known as the sense of touch, is a crucial and commonly used sense in everyday life. Somatosensory deficit is a common symptom in both peripheral and central neurological diseases and insults [1], such as lesions of sensory receptors, lesions of the peripheral nerves, or impairments to haptic representation in the central nervous system (specifically in the primary sensory cortex, such as stroke) [2,3]. Therefore, tactile spatial acuity (TSA) is commonly measured to evaluate somatosensory function [4,5] and yields important information for diagnostic, functional, and prognostic purposes.

Tactile spatial and directional sensitivity on a body part is mainly determined by two factors, the density of peripheral mechanoreceptors and their receptive field properties [6,7]. Several assessments have been developed to objectively measure tactile spatial resolution in various physiological or pathological conditions. Two-point discrimination, a tactile spatial discrimination task, is the most widely used for this purpose [6]. In recent decades, the utility of two-point discrimination has come under scrutiny because it can only measure at the threshold of a just-noticeable difference instead of the limit of spatial resolution, and most importantly, it tends to yield inconsistent results because it cannot control nonspatial cues [5,6]. The JVP dome, a grating dome that assesses sensory capacity in grating orientation discrimination [2], is regarded as the standard method used to qualify the tactile threshold for the spatial resolution [3]. Grating orientation sensitivity is considered suitable for assessing spatial acuity because it is affected by the density of innervation and varies with the somatosensory function of the fingerpad [8]. Nevertheless, the use of the JVP dome requires considerable skill and the indentation depth on the skin cannot be accurately controlled because it is delivered by the examiner’s hand, making it necessary to develop a fully automatic method that allows the tactile orientation discrimination task to be performed.

The tactile orientation discrimination task has been performed by healthy people [5,6], patients with neurological disorders [3,9,10], and blind Braille readers [11,12]. Studies have reported that the TSA of blind Braille readers is superior to that of healthy people [11,12], whereas the TSA of patients with neurological disorders is worse [3,9,10]. Moreover, TSA can improve with training [11,13]. The tactile orientation discrimination task has also been performed by children [14] and older adults [15,16], indicating that older age is correlated with a decline in tactile spatial resolution. 

The miniature tactile stimulator (MTS) was developed by our group for performing automatic tactile stimulations that can cover a variety of movement directions and speeds; these simulations are used to characterize the human ability to perceive shape and motion by touch [17]. MTS consists of three independently controlled micromotors, providing three degrees of freedom, which are the grating ball’s direction of movement, speed of rotation, and depth of vertical indentation on the skin [17]. We successfully demonstrated the functionality of the MTS for measuring the performance of tactile motion discrimination in healthy participants [17,18,19], and the MTS thus offers unique potential as a fully automatic and standardized measurement of tactile acuity.

In the present study, we developed a method where MTS is used for an orientation discrimination task; this method can be applied to the measurement of the severity of somatosensory deficits. Through MTS, motion stimuli were applied to the participant’s fingerpad, and the participant reported the received orientation of the moving grating ball. We describe how the measurement was performed during the training and testing phases. Specifically, the orientation discrimination task was performed at a variety of motion speeds to characterize how much motion speed affected tactile acuity. We demonstrated that MTS-based orientation discrimination tasks can be effectively applied to evaluate tactile acuity in healthy participants.

## 2. Protocol

### 2.1. Participants

This study and the experimental protocol were reviewed and approved by the institutional review board (IRB) of the Chang Gung Medical Foundation (IRB no.: 202001628B0A3). All of the methods were performed in accordance with the regulations of the Taiwan Human Subjects Research Act and the guidelines of the Declaration of Helsinki 1975. The details of the study and the procedures were clearly explained to each participant.Sixteen healthy participants were recruited for the main experiment, and written informed consent was obtained from all participants (Figure 1).The inclusion criteria were as follows: (1) aged between 40 and 65 years old, and (2) normal cognitive function as indicated on a judgment, orientation, memory, abstract thinking, and calculation (JOMAC) scale.The exclusion criteria were as follows: (1) sensory loss (anesthesia or hypoesthesia) or sensation change (paresthesia or hyperesthesia), (2) lesion at peripheral or central nervous systems, (3) entrapment neuropathy, (4) alcoholism or history of alcoholism, (5) rheumatoid disease, (6) complex regional pain syndrome, (7) hypothyroidism, (8) fibromyalgia, (9) end stage renal disease (ESRD), (10) inability to perform the button press to report perceived orientation or direction, and (11) obvious calluses or wounds noted on the index fingertip.

### 2.2. Experimental Setup

5.The MTS system was set up on a desk and connected to a PC through RS-232 ports (Figure 2A).6.The MTS system comprised a three-motor controller for manipulating the indentation depth, directions of motion, and rotation speeds of the grating ball; a finger support surface for the participant to place their finger on; and a human–machine interface for controlling the MTS [17].7.MTS consisted of three degrees of freedom (DoF) (one of which is illustrated in Figure 2B): rotation for producing motion, vertical excursion for controlling the depth of indentation into the skin, and arm orientation for controlling the direction of motion. Each DoF was controlled by a single motor: two DC motors controlled the rotational motion and direction of motion, while one DC motor controlled the depth of indentation.8.The user interface (UI) included several parts: the first part was for registering information on the participants’ general characteristics; the second part was for setting up the three control parameters (indentation depth, motion direction, and rotation speed) of the grating ball; the third part was for recording the participant’ responses in the orientation discrimination task, where they selected the corresponding vertically and horizontally oriented pictures on the screen; and the fourth part was for analyzing the results of the task immediately after each training or testing session.9.The grating ball, engraved with 2 mm sine-wave gratings with a peak-to-peak amplitude of 0.5 mm, was installed on the stimulator (Figure 2C).10.On each trial, the grating ball first rotated the orientation of the grating ball (Figure 2B), and then rotated the grating ball to reach the defined surface scanning speed. Next, the rotating grating ball compressed the fingerpad with a fixed indentation depth of 1 mm for 1 s to present the tactile stimulation. Finally, the rotated grating ball moved upward, left the skin, and then stopped rotating.11.Headphones and a blindfold were prepared, which prevented the participants from seeing the grating orientation and hearing the motor noise of the MTS.12.Information on the participants’ basic characteristics was registered, and tactile stimuli parameters were set into the control program via the UI.13.The participants’ index finger was secured inside the finger holder using Velcro. The forearm and hand were positioned palm side up on the forearm supporter and the base of the MTS.14.The grating ball was adjusted to the surface of the participants’ finger, to make sure that the indentation of the tactile stimuli during tests was precise (Figure 2B).

**Figure 2 sensors-23-01844-f002:**
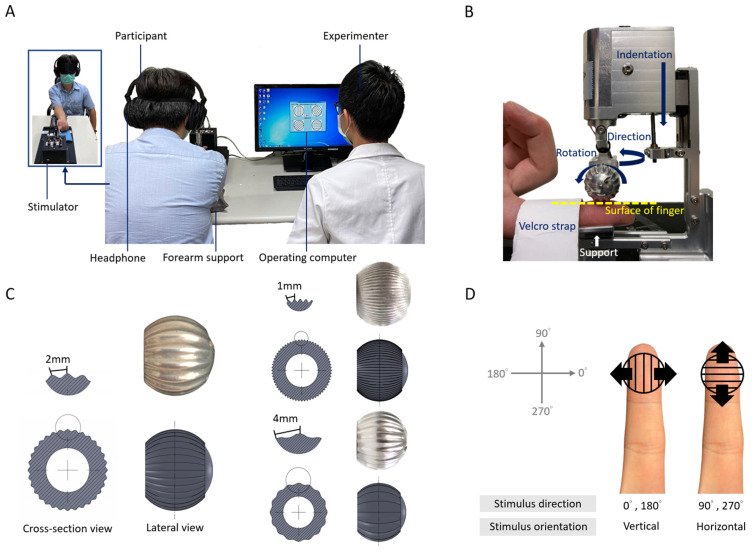
The miniature tactile stimulator (MTS) and experimental setup. (**A**) Setup of test experiment. (**B**) Three degrees of freedom (indentation, direction, and rotation) provided by MTS and the presentation of the rotating grating ball on the fingerpad. (**C**) The cross-sectional and lateral views of the 2 mm sine-wave grating ball (left), and which of the 1mm and 4 mm sine-wave grating balls (right upper and lower, respectively) for additional stimulations. (**D**) The definition of the stimulation directions and their corresponding orientations.

#### 2.2.1. Instruction Session

15.The participants were informed of the procedure, safety precautions, and purpose of the experiment. The participants were then asked to sign the written informed consent form.16.The experimenter measured TSA using the MTS and comprised three sessions: eye-opened training, eye-closed training, and testing.17.The MTS performed tactile stimulations on the index fingertip of the participant’s right hand. The procedure was noninvasive and painless, and the total duration was approximately 30 min.18.In the main experiment, in each trial, the participants were presented with one of the stimulus conditions from a factorial design of 16 combinations (4 directions × 4 speeds) in a pseudorandom order. Specifically, the directions were 0°, 90°, 180°, or 270° and the surface moving speeds were 5, 10, 40, or 160 mm/s. The tactile stimulus had an indentation depth of 1 mm, and each stimulus lasted for 1s.19.The reference frame was defined with respect to the forearm when the patients were in a neutral position. Directions of 0° and 180° indicated that the grating ball was moving rightward and leftward, respectively, and thus yielded a vertical orientation as the grating orientation was orthogonal to the scanning direction. Similarly, 90° and 270° indicated that the grating ball was moving distally and proximately, respectively, and thus yielded a horizontal orientation (Figure 2D). Although the participants were presented with one of the four directions, the participants’ task was to report the perceived grating orientation (horizontal or vertical).

#### 2.2.2. Eye-Opened Training Session

20.The participants’ fingers were fixed at the appropriate location inside the finger holder (Please see numbers 9 and 10 in Section 2.2).21.The aim of this task was to confirm that the participants understood the task structure.22.The participants were instructed to visually inspect the orientation of the grating engraved on the stimulation ball during stimulation.23.The tactile stimulation was identical to that described in numbers 4 and 5 in Section 2.2.1, with a fixed surface moving speed of 40 mm/s. We employed a total of four combinations (4 directions × 1 speed) and repeated each combination three times, yielding a total of 12 trials (4 directions × 1 speed × 3 repetitions). In each trial, the participants were thus presented with one of the vertical or horizontal orientations (see number 5 in Section 2.2.1). As a two-alternative forced choice design, the participants verbally reported the perceived orientation after each tactile stimulus.24.Based on our inclusion criteria of cognitive function, we expected that all participants would meet the passing criteria during training.

#### 2.2.3. Eye-Closed Training Session

25.This was a 24-trial task where the participants’ eyes remained closed.26.The participants were asked to wear a blindfold and headphones that played white noise so that they could not see or hear the stimuli.27.The aim of the eye-closed training session was to ensure that the participants could perform the task under conditions analogous to those of the testing session.28.Each stimulation was performed as described in number 4 and 5 of Section 2.2.1, with a fixed surface moving speed of 40 mm/s. We employed a total of four combinations (4 directions × 1 speed) and each of the combinations was repeated six times (4 directions × 1 speed × 6 repetitions = 24 trials).29.The accuracy, quantified by the probability that the participants would give a correct answer, was computed from the aforementioned 24 trials. If the participants’ accuracy was ≥75%, we proceeded to the testing session; if not, we repeated the eye-opened training (number 3 and 4 in Section 2.2.2) to make sure they understood the task.30.The participants were excluded if their accuracy did not reach 100% during the second round of eye-opened training.

#### 2.2.4. Testing Session

31.The setup of the testing session was identical to that in the eye-closed training session (see number 2 in Section 2.2.3), and the stimulus protocol was identical to that delineated in number 4 in Section 2.2.1.32.The testing session involved three blocks, and each block had a factorial design of 32 combinations (4 directions × 4 speeds × 2 repetitions), for a total of 96 trials.33.The results of each patient in the trial were analyzed.34.In order to investigate the effect of motion of rotation with the grating edge on the fingerpad, some of the participants (8 of 16) also received additional stimulations by using 1 mm and 4 mm sine-wave grating balls (Figure 2C).35.The procedure for the stimulation using 1mm and 4mm grating balls was the same as that of the 2 mm grating ball.

### 2.3. Usability Test

36.The system usability scale (SUS) [20,21] questionnaire was applied to examine the usability of MTS in participants who had no experience operating the device.37.The recruited participants were briefly instructed with our MTS-based protocol.38.Our examiner firstly operated MTS to test the participants’ tactile acuity using the protocol.39.Next, the participants would try to operate MTS and test the examiner’s tactile acuity independently.40.Finally, the participants completed the SUS questionnaire.

### 2.4. Data Analysis

41.The statistical analyses were performed using Statistical Program for Social Sciences (SPSS). The Wilcoxon signed-rank test was applied to compare the inter-orientation accuracy between the vertical and horizontal orientations. The Friedman test was applied to compare the accuracy between speeds, and if the results were significant, the Wilcoxon signed-rank test was used as the post hoc test. The repeated measure ANOVA was applied to the full 3 wavelength × 4 speed model, in which factors are wavelength (1, 2, and 4 mm sine-wave grating ball) and speed (5, 10, 40, and 160 mm/s). The confidence interval was corrected using the Bonferroni method.42.All data are presented in terms of the mean ± standard error of the mean (SEM) with 95% confidence intervals.43.The SUS questionnaire consisted of 10 statements with five positive and five negative statements and each question had a five-point scale (strongly disagree = 1 point and strongly agree = 5 points).44.For analysis of SUS, the odd-numbered questions, Q1, 3, 5, 7, and 9, were positive questions, and the recorded scores were: original score − 1.45.Even numbered questions, Q2, 4, 6, 8, and 10, were negative questions, and the recorded scores were: 5 − original scores.46.The recorded scores from the ten questions were summed up and then multiplied by 2.5 to yield the total SUS score.47.The total SUS score ranged from 0 to 100. When it came to an acceptable SUS score, products were at least passable when SUS scores were over 70. Good products could score in the mid-70s to low-80s. Excellent products would score better than 85. Products with scores less than 70 needed to be considered for scrutiny and improvement [22].

## 3. Results

Sixteen healthy participants (eight men and eight women, aged 48.50 ± 1.50 years) were enrolled (Table 1). We tested their right-hand tactile accuracies, but two of them were excluded due to carpal tunnel syndrome in at least one of their hands. All of the remaining participants completed the training and formal experiments and no adverse events were reported.

In the eye-opened training session, all of the participants achieved 100% accuracy (*n* = 14). Data from the training session indicated that the participants fully understood the experimental protocol. This supported the effectiveness of the instruction to the participants. In the eye-closed training session, eleven of the fourteen participants passed the threshold (threshold = 75%, Table 2). Although participants #3, #13, and #16 initially failed the eye-closed training (accuracy = 62.5%, 45.83%, and 58.33% respectively), these participants passed after receiving an additional eye-opened training session (accuracy = 100%).

The demographic data in the testing session are shown in Table 3. The mean accuracy was 89.36% ± 1.12% (*n* = 14), which exceeded the threshold of 75%. The results also demonstrated that accuracy in the vertical (87.35 ± 1.55%) and horizontal (91.37 ± 1.49%) orientations did not significantly differ (*p* = 0.08, Figure 3A).

Subsequently, we characterized the effect of motion speed on the performance of orientation identification. The results demonstrated that tactile accuracy differed across speeds (*p* < 0.001, Friedman test), and the post hoc test revealed that accuracy at the highest speed (160 mm/s) was significantly inferior to that in each of the other three speeds (5, 10, and 40 mm/s, Figure 3B). This finding indicates that scanning speed might affect the participants’ performance, which could be applied for assessing tactile acuity under various conditions.

We examined whether the wavelength of the grating affects the accuracy of orientation discrimination, a phenomenon that could support that the percept was mainly mediated by spatial information presented by the grating rather than the shear force. To this end, 8 of the 14 participants were presented with sinusoid grating balls with wavelengths of 1, 2, or 4 mm. The results showed that accuracy was significantly modulated by wavelength (*p* = 0.006) and the post hoc analysis showed that the 1mm grating had a worse accuracy compared with the 4mm grating (Figure 4A, *p* = 0.019). The performance was thus better with a wider wavelength of grating.

In addition, we examined whether the scanning speed affected the accuracy of orientation discrimination. The results showed that accuracy was significantly modulated by motion speed (whole ANOVA model here *p* = 0.009) and the post hoc analysis showed that 160 mm/s had a worse accuracy compared with 5 and 160 mm/s (Figure 4B, *p* = 0.042), indicating that the performance decreased at extremely high speeds. Moreover, there was no interaction effect between the wavelength and scanning speed (Figure 4B, *p* = 0.733), indicating that the aforementioned wavelength effect was independent of the scanning speed.

Finally, we evaluated the usability of the MTS protocol using the SUS questionnaire in seven naïve examiners (*n* = 7) (Table 4). The results demonstrated that the overall SUS score was 88.57 ± 3.61, indicating that the usability of the present protocol was excellent (the overall SUS score >85) [22]. The recorded scores in all items were ≥3 in all questions, except item Q4 (Table 5), “I required technical assistance to use the miniature tactile stimulator”, in which the score was only 2.71 ± 0.52, suggesting that the participants might need assistance when operating the device.

In summary, our results demonstrated that the novel procedure for tactile stimulation could be applied as a psychophysical assessment for examining the tactile acuity of healthy individuals.

## 4. Discussion

MTS is a robotic tactile stimulator that could be applied for tactile function capability assessments [17,18,23]; it is novel because a standard and feasible method has yet to be established. In the present study, we developed an MTS-based program for tactile acuity evaluation. The results demonstrated the following: (1) the examining process could be easily performed by examiners, and all of the participants passed both training sessions with accuracy above the predefined threshold, a finding that supports the utility of the pre-test training; (2) neither discomfort nor adverse events were reported by the participants, suggesting the acceptability of this designed tactile measurement protocol; (3) all participants had above-threshold accuracy (accuracy ≥ 75%) when identifying the directions of motion at lower speeds (5, 10, and 40 mm/s), indicating that most healthy participants could perform the present protocol; and (4) the accuracy of orientation identification decreased at higher speeds, suggesting that higher speed (160 mm/s) conditions could be more challenging for tactile testing. In summary, the present study developed an evaluation protocol that could apply MTS for the automatic testing of tactile acuity.

This is the first protocol designed for the assessment of tactile function using an automatic robotic system with MTS. Because we aimed to apply this program to patients with somatosensory deficits, the protocol was designed with an adjustable degree of difficulty that can discriminate between various severities of somatosensory impairment. Considering user comfort in the testing program, we modified the body postures through which the participants received the assessment. In addition, the duration of the entire examination procedure was optimized and was shown to be tolerable by all participants.

Tactile orientation discrimination tasks have been applied to distinct body parts, such as the lip, tongue, and fingers. The levels of TSA sequenced were highest to lowest in the lip, tongue, index or ring finger, and little finger, respectively [2,4,24]. In the present study, we chose the index finger as our measuring target for its advantages of fine TSA and easy manipulation compared with the lip or tongue.

An important question is whether the participant relied on the stretch force induced by the scanning movement rather than the spatial–temporal information presented by the grating orientation [19,25]. Indeed, in the present setup, the shear force was always orthogonal to the grating orientation, so that it was possible that the participants used the direction of shear force to infer the grating orientation. Using sine-wave grating with a variety of wavelengths, we observed a strong effect of wavelength on the accuracy, a finding that is reminiscence of the effect of grid width on static orientation judgement [5,26]. Furthermore, speed and wavelength are two independent factors that influence accuracy, suggesting that wavelength as a spatial factor has a distinct mechanism to affect performance and this mechanism is not affected by speed. To this end, these findings support that the participants’ performance was mainly determined by spatial–temporal cues.

For usability, the results showed that most of the SUS items had high scores, indicating that a naïve examiner could operate the MTS well after brief instruction and training. However, we found the lowest score in item Q4, suggesting that non-experienced operators might require help from others. For example, the operator sometimes hesitated to select the settings on the operational interface software and needed assistance from the experienced operator. In our opinion, protocols that simplified and optimized operative steps are expected to improve the users’ satisfaction and its clinical usability.

The MTS system and present study have several limitations. First, the sample size was small because this was a pilot study for assessing the feasibility and acceptability of the MTS-based program for measuring tactile acuity. Second, it was difficult to apply this tactile function assessment instrument to some parts of the body, such as the lip and tongue, because more adjustment was required to apply the MTS on other body parts. Third, although the protocol was optimized in the present study, it still took up to 30 min for each participant to complete the whole process. Fourth, it is possible that the subjects still had minute finger movements when the fingers were placed on the finger holder during the experiment. Without continuous monitoring of finger position, it was hard to make sure the participants remained in their assigned position throughout the experiment.

In addition, participants may have still been able to perceive the direction of motion, relying on the spatiotemporal cues provided by the miniature ball’s rotation and stretch cues—the rotation exerted on the skin surface. Indeed, tactile direction discrimination is the ability to recognize the tactile motion direction of an object moving across the skin [27,28]. Sensing the direction of a tactile stimulus relies on spatiotemporal cues and skin stretch through the simultaneous processing of information signaled by large myelinated afferent nerve fibers and dorsal column pathways [29,30]. Tactile direction judgments are sensitive to the direction of skin stretch, whereas low friction stimuli with minimal skin stretch, such as the rolling ball, provide only spatiotemporal cues (successive positions cues) for the direction of motion [31,32,33]. Furthermore, SA1 receptors are at least ten times more sensitive to moving than to static stimuli [25].

The MTS system is an automatic, noninvasive, and quantitative method for assessing tactile acuity. Our study demonstrated that this novel MTS-based protocol is safe, accessible, tolerable, and feasible as a simple standardized measurement of tactile acuity. We will conduct further studies where we will apply this program to evaluate somatosensory deficits in patients with peripheral neuropathies or central neurological disorders.

## Figures and Tables

**Figure 1 sensors-23-01844-f001:**
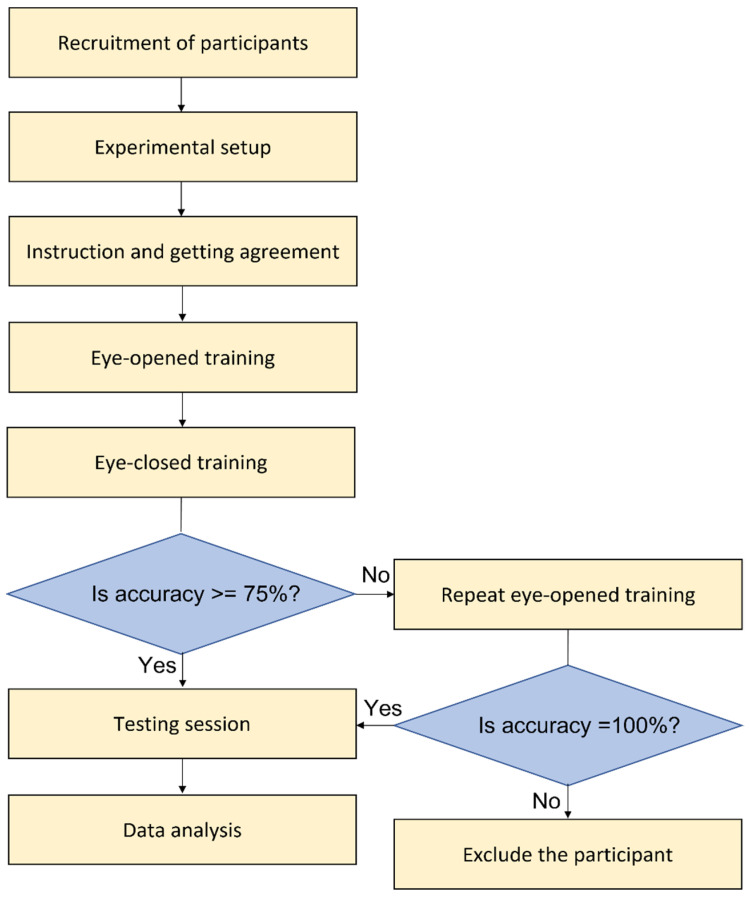
Flowchart of the experimental procedure.

**Figure 3 sensors-23-01844-f003:**
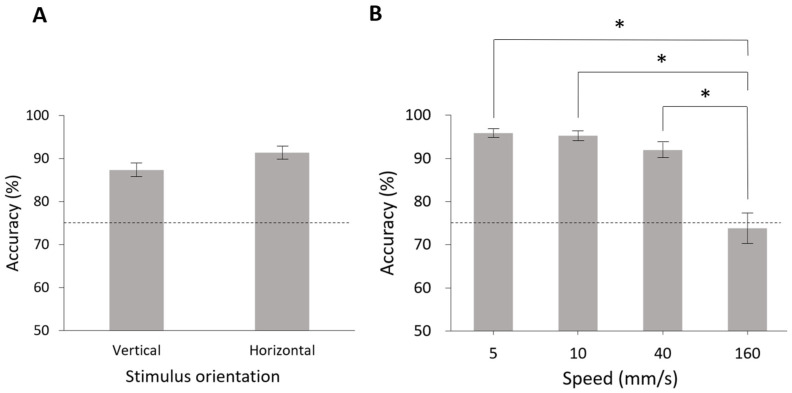
Performance results in terms of accuracy of orientation identification in different stimulus orientations and speeds. (**A**) Comparison of accuracy for the vertical and horizontal orientations revealed no significant difference (*p* = 0.08). The accuracy for stimulus orientations of vertical and horizontal were 87.35 ± 1.55% and 91.37 ± 1.49%, respectively. The error bars indicate SEM. (**B**) Comparison of accuracy for speeds of 5, 10, 40, and 160 mm/s revealed that tactile accuracy differed across 160 mm/s and the other speeds (*p* < 0.001, Friedman test). The accuracy for speeds of 5, 10, 40, and 160 mm/s were 95.83 ± 0.98%, 95.24 ± 1.06%, 91.96 ± 1.77%, and 73.81 ± 3.48%, respectively, * indicates *p* < 0.05 in post hoc comparisons.

**Figure 4 sensors-23-01844-f004:**
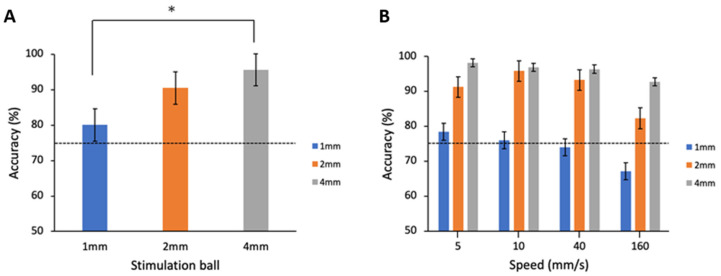
Performance results of 1, 2, and 4 mm sine-wave grating ball in terms of accuracy of orientation identification in different stimulus orientations and speeds. (**A**) Comparison of the accuracy of 1, 2, and 4 mm sine-wave grating ball stimulation revealed a significant difference (*p* = 0.019). The accuracy of 1, 2, and 4 mm sine-wave grating ball stimulation is 77.67 ± 5.14%, 89.88 ± 1.99%, and 95.93 ± 1.45%, respectively. The error bars indicate SEM. (**B**) Comparison of the accuracy for speeds of 5, 10, 40, and 160 mm/s of 1, 2, and 4 mm sine-wave grating ball stimulation revealed that tactile accuracy differed across 5 mm/s and 160 mm/s (*p* < 0.001, repeated measure ANOVA test). The accuracy for speeds of 5, 10, 40, and 160 mm/s are 96.87 ± 4.02%, 97.57 ± 3.30%, 94.79 ± 5.06%, and 73.26 ± 12.24%, respectively, * indicates *p* < 0.05 in post hoc comparisons.

**Table 1 sensors-23-01844-t001:** Participant characteristics.

Participant	Age (Years)	Sex	Test Hand	Education Level
#1	55	Female	Right	University
#2	49	Female	Excluded *	Junior high school
#3	51	Male	Right	University
#4	40	Male	Right	University
#5	49	Female	Excluded *	Senior high school
#6	56	Male	Right	Elementary school
#7	54	Female	Right	Senior high school
#8	55	Female	Right	University
#9	50	Male	Right	University
#10	49	Male	Right	University
#11	43	Female	Right	University
#12	41	Male	Right	University
#13	49	Male	Right	University
#14	45	Female	Right	University
#15	41	Male	Right	University
#16	50	Female	Right	University

* These two participants were excluded due to right-hand carpal tunnel syndrome.

**Table 2 sensors-23-01844-t002:** Accuracy in training sessions.

Participant	Accuracy in Training Sessions (%)			Pass Training Criteria?
	Eye-Opened Training (1st)	Eye-Closed Training	Eye-Opened Training (2nd)	
#1	100	95.8	-	Yes
#3	100	62.5	100	Yes
#4	100	83.3	-	Yes
#6	100	87.5	-	Yes
#7	100	100	-	Yes
#8	100	95.8	-	Yes
#9	100	87.5	-	Yes
#10	100	95.8	-	Yes
#11	100	91.7	-	Yes
#12	100	83.3	-	Yes
#13	100	45.8	100	Yes
#14	100	83.3	-	Yes
#15	100	83.3	-	Yes
#16	100	58.3	100	Yes

**Table 3 sensors-23-01844-t003:** Accuracy of the testing session.

Participant	Accuracy in Different Speeds (%)				Accuracy in Different Orientations (%)		Total Accuracy (%)
	5 mm/s	10 mm/s	40 mm/s	160 mm/s	Vertical	Horizontal	
#1	95.8	95.8	87.5	70.8	89.6	85.4	87.5
#3	95.8	95.8	91.7	58.3	79.2	91.7	85.4
#4	87.5	91.7	100	79.2	85.4	95.8	89.6
#6	91.7	100	95.8	95.8	95.8	95.8	95.8
#7	100	100	100	58.3	79.2	100	89.6
#8	100	100	91.7	75	93.8	89.6	91.7
#9	95.8	100	91.7	58.3	87.5	85.4	86.5
#10	100	95.8	100	87.5	93.8	97.9	95.8
#11	91.7	91.7	87.5	87.5	89.6	89.6	89.6
#12	95.8	95.8	91.7	79.2	85.4	95.8	90.6
#13	95.8	91.7	75	54.2	77.1	89.6	79.2
#14	95.8	87.5	87.5	83.3	81.3	87.5	88.5
#15	95.8	95.8	91.7	79.2	85.4	95.8	90.6
#16	100	91.7	95.8	66.7	91.7	87.5	88.5

**Table 4 sensors-23-01844-t004:** Participant characteristics for SUS.

Participant	Age (Years)	Sex	EDUCATIONAL LEVEL
#1	25	Male	Senior high school *
#2	31	Male	University
#3	25	Male	Senior high school *
#4	27	Female	University
#5	41	Female	University
#6	32	Male	University
#7	32	Male	University

* The participants were undergraduate.

**Table 5 sensors-23-01844-t005:** SUS score obtained from the operators.

Item	Content	Score
		All Subjects (*n* = 7)
Q1	I would like to use the miniature tactile stimulator often	3.57 ± 0.20
Q2	I think the miniature tactile stimulator is complex to use	3.43 ± 0.20
Q3	I think the miniature tactile stimulator is easy to use	4.00 ± 0.00
Q4	I required technical assistance to use the miniature tactile stimulator	2.71 ± 0.52
Q5	I think the functionalities of the miniature tactile stimulator are well integrated	3.71 ± 0.18
Q6	I think the functionalities of the miniature tactile stimulator are not consistent	3.43 ± 0.20
Q7	I think most users can quickly learn to use the miniature tactile stimulator	3.86 ± 0.14
Q8	I think most users have difficulties learning to use the miniature tactile stimulator	3.86 ± 0.14
Q9	I am confident when using the miniature tactile stimulator	3.43 ± 0.30
Q10	I need to learn more background information of the miniature tactile stimulator before use	3.43 ± 0.20

## Data Availability

The data that support the findings of this study are available from the corresponding author upon reasonable request.

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
