# Peer review of "A Novel Tactile Function Assessment Using a Miniature Tactile Stimulator"

_sensors, 2023, doi:10.3390/s23041844_

Round 1
Reviewer 1 Report (New Reviewer)
Major points:
1. There is no comparison to norm data, which is important for validating the new device.
2. The rationale of including moving speed and direction as factors is not explained. Why not using static stimuli?
3. The authors should consider analyzing the data using signal-detection method to isolate response bias and sensitivity.
4. The sample size is small. The authors should justify the choice of sample size in terms of the statistical power.
Minor points:
p.3
1.1: How many participants were recruited/tested?
p.4
3.1.2: The “experimenter” measured TSA […]
3.1.4: Should it be 32 combinations (2 grating orientations * 4 directions * 4 speeds)?
3.2.4 & 3.3.4: Same as 3.1.4, what about the factor of grating orientation?
3.2.5: What is the rationale of using 75% as the passing criterion?
p.5
3.3.5-3.3.6: Are they retested on the eye-closed task?
3.4.4: How many participants were also tested on 1mm and 4mm?
5.1: Why perform a separate Friedman test on speed if a full-factorial ANOVA was performed? And is the repeated-measures ANOVA performed only on those participants who were tested on all wave lengths?
p.8
The sample size for the ANOVA analysis is rather small (n=8). The authors should justify their sample size in terms of statistical power.
Author Response
Please see the attachment.

Reviewer 2 Report (New Reviewer)
The present study develops a novel method applying a robotic tactile stimulator to automatically measure tactile acuity that comprises eye-opened, eye-closed training, and testing sessions. Healthy participants judge the orientation of a rotating grating ball presented on their index fingerpads in a two-alternative forced-choice task. A variable rotation speed of 5, 10, 40,or 160 mm/s was used for the tactile measurement at a variety of difficulties. All participants met the passing criteria for the training experiment. Performance in orientation identification, quantified by the proportion of trials with correct answers, differed across scanning directions, with the highest rotation speed (160mm/s) having the worst performance. Accuracy did not differ between vertical
and horizontal orientations. Theirs results demonstrated the utility of the pretest training protocol and the functionality of the developed procedure for tactile acuity assessment. The novel protocol performed well when applied to the participants. Future studies will be conducted to apply this method to patients with impairment of light touch.
The detailed review comments are as follows:
1. It is suggested that the author supplement the comparison between this study and previous studies to highlight the innovation of this study.
2. For the grating ball in the micro tactile stimulator, if it is replaced by a ball with a smooth surface, whether the tensile force caused by scanning motion can be reduced, please supplement the control experiment.
3. As for the SUS questionnaire, the number of participants is not universal, please increase the number of people who fill in the questionnaire.
4. Please explain that the accuracy of the sine wave grating ball with the speed of 10mm/s and 2mm in Fig. 4b is higher than that of the grating ball with other speeds.
5. The participants in the experiment are no more than 60 years old. It is suggested that the participants selected by the author are universal.
6. For the finger experiment, if there are calluses or wounds on the finger, whether the tactile test results will be affected, please add relevant instructions.
7. It is suggested that the author should add relevant tests and test results applied to patients with somatosensory deficits.
Author Response
Please see the attachment.

Reviewer 3 Report (New Reviewer)
The topic in this paper is interesting and can be acceptable.
Round 2
Reviewer 1 Report (New Reviewer)
The authors have addressed my comments.
Reviewer 2 Report (New Reviewer)
Authors have improved the manuscript in the revision considering the reviewers' comments. I support the publication of this manuscript with no further comments on its contents.
This manuscript is a resubmission of an earlier submission. The following is a list of the peer review reports and author responses from that submission.
Round 1
Reviewer 1 Report
In the past decays, several measurement methods for tactile acuity have been developed, however, many defects were found in practical application and evaluation. In this research work, the authors and his co-authors designed and developed a whole new method to automatically measure tactile acuity, which obtained a satisfactory and convincing result. However, some issue still need to dressed before publishing the work in the sensor and I also recommend is to be publish in this journal.
- In the whole protocol, the authors only recruited six healthy participants, whether it need to expand the population to reduce the random error, how about the age range, physical condition, please give out the detailed explanation.
- The author should describe in detail how the MTS works, whether other instruments can replace its main functions, and whether the resulting functions are representative.
- There are some grammatical problems for the author to check in detail.
Reviewer 2 Report
The present manuscript covers an interesting topic, the quantitative assessment of tactile function, particularly in relation to clinical populations who experience problems with touch. This is currently assessed with rudimentary methods in the clinic, and as the authors quite rightly point out that this could very much be improved in order to aid chances of better diagnosing somatosensory disturbances a d informing diagnoses. However, there are some major issues with the manuscript and the study that need addressing.
A major problem throughout the manuscript is that the authors commonly refer to tactile acuity in relation to their study. Tactile acuity is clearly measured by the two point discrimination test, or indentation with the JVP dome cited in the manuscript. There are problems with these methods as the authors point out, but these are in essence pure spatial stimuli. Perception of these, and performance in the psychophysical task is based upon the perception of the spatial pressure distribution on the skin for detection of distinct points or the stimulus pattern. Unless specific evidence is presented to the contrary , the stimulator the authors have used seems likely to induce some tangential skin stretch clues about movement direction, which human subjects have a remarkable capacity for (several studies by Olausson and colleagues on this topic are cited, and Olausson 1998 gives a particularly detailed illustration of this). Without mentioning the material the stimulus ball is made from, or the ball width/curvature, the extent of skin stretch is hard to estimate. This will also affect the number of grating periods which are in contact with the skin. If the authors could demonstrate no lateral skin stretch is induced by the stimulator (given the perceptual accuracy for skin stretch can be as low as 0.13mm, Olausson 1998), this would provide a more convincing argument about what the perceptual cues might be. Modulating ball groove width as the authors mention might be a possibility of indictaing that performance of this task is dependent on acuity. This at least needs discussion, but quite possibly would need a condition where participants are instructed to detect direction using a stimulator with similar frictional cues, but a less oragnised spatial arrangement.
The introduction generally seems sufficient to outline the scope and importance of the studied topic.]
The study designs and the conclusions that can be made from this do not necessarily seem appropriate to address the question of whether the stimulator provides an adequate, useful or straightforward assessment of tactile function, and thus whether this presents a significant improvement over existing methods of assessment. The results are essentially, can subjects perform the task to the arbitrarily decided performance level, and if the difficulty of the task be adapted by changing the speed of stimulation. Very few participants have been included in the study and the conclusion that ‘most healthy participants can perform the present protocol’ (page 8) is based on n=6. This is not a sufficient sample to confirm a hypothesis about population performance. This is mentioned in the discussion (page 8) because this was a ‘pilot study for feasibility and acceptability’, but this does not seem like a reasonable justification for and inadequate sample size in a published article.
Although the authors claim to have ‘demonstrated the examining process could be easily performed by examiners’, if this is based upon the researchers who have designed and have experience with the apparatus perfoming the procedure, this does not seem like an appropriate conclusion. If the procedure could be performed by researchers/examiners naïve to the apparatus, that would support this conclusion. The ability to perform the task, and the ease of use of the equipment are in essence the major claims of the study, and I don’t think are supported by the study design.
Another claim that has not been substantiated is the necessity/usefulness of the pre-examination training. A no visual training condition would need to be used to support this. Performing a pre visual training condition would be a useful way to evaluate this. It may be that a with vision training is only necessary if there is poor task performance, which would enhance the speed of use, over performing a separate pre-examination, unless a need for this is demonstrated.
There is also the further fairly major problem with the study design, the inclusion of two participants with a neurological condition affecting tactile function (carpal tunnel syndrome). This is somewhat concerning, and that these subjects may well not fill the stated criteria mentioned for neurological impairments. Carpal tunnel syndrome commonly develops bilaterally, and even if this has only been a unilateral diagnosis. I think at mentioning that these subjects had normal tactile function in their left (unaffected) hands should be necessary, or another test of tactile function acuity, discrimination, or at least a comment on the relative performance of these subjects in the task. All participants are additionally all on the border of what would be considered to be an ’elderly’ population (generally starting at 60 years) this is not mentioned at all in the text. There may well be some extent of decline in tactile function at this point, and the application of the task to young healthy subjects may not necessarily produce similar results.
As well as lack of description of the stimulator (as mentioned above in the context of skin stretch), the methods omit important details of the stimulation which are needed for interpretation. The indentation of the stimulator is not described. When the orientation of the stimulator is changed, this must leave the skin and be re-indented. It should be clearly mentioned if this is a static indentation pre stimulator movement, or if the stimulator is indented whilst moving. This cannot be understood currently. If pre movement indentation is performed., a critical control would be to have a no movement condition, to examine whether cues from the initial indentation might inform the perceptual judgements. The timing and speed to stimulator contact should be clearly described.
An improvement claimed over existing equipment is the precision/ease of operating. However, subject movement is frequently hard to control. A reliable, repeatable and ‘precise’ 2mm indentation will only be achieved by precise manual positioning by the experimenter and a complete lack of suject movement. This should be discussed as a drawback of the stimulator, and further improvements would be to better control and automate this part of the procedure. Force detection or skin impedence measurements may be used to better control stimulus contact and compensate for small movements which would not be practical to so highly constrain at other body sites, or under a clinical conditions.
Overall, although the topic is interesting, to make the study of wider interest I believe a larger sample size is needed, the ease of use for non experienced operators should be demonstrated, and some element of acuity should be manipulated in order to make the claims made in the study (period of the stimuli, skin areas with different tactile acuities). Therefore, I believe further experiments would be required to substantiate the claims and demonstrate utility of the device.
There are additional fairly minor problems with the language, but I will not detail these for a major revision